# Continuous In Situ Measurement of Dissolved Methane in Lake Kivu Using a Membrane Inlet Laser Spectrometer

Roberto Grilli[1], François Darchambeau[2], Jérôme Chappellaz[1], Ange Mugisha[3], Jack Triest[4] and Augusta Umutoni[3]

[1] CNRS, Univ. Grenoble Alpes, IRD, Grenoble INP, IGE, F-38000 Grenoble, France
[2] KivuWatt Ltd., Kigali, Rwanda and Chemical Oceanography Unit, Université de Liège, Belgium
[3] Lake Kivu Management Program LKMP, Gisenyi, Rwanda
[4] KM Contros, Kongsberg Maritime, Kiel, Germany

*Correspondence to*: Roberto Grilli (roberto.grilli@cnrs.fr)

**Abstract.** We report the first high resolution continuous profile of dissolved methane in the shallow water of lake Kivu, Rwanda. The measurements were performed using an in situ dissolved gas sensor, called Sub-Ocean, based on a patented membrane-based extraction technique coupled with a highly sensitive optical spectrometer. The sensor was originally designed for ocean settings, but both the spectrometer and the extraction system were modified to extend the dynamical range up to six orders of magnitude with respect to the original prototype (from nmol $L^{-1}$ to mmol $L^{-1}$ detection) to fit the range of concentrations at lake Kivu. The accuracy of the instrument was estimated to ±22% ($2\sigma$) from the standard deviation of eight profiles at 80 m of depth, corresponding to ±0.112 mBar of $CH_4$ in water or ±160 nmol $L^{-1}$ at 25°C and 1 atm. The instrument was able to continuously profile the top 150 m of the water column within only 25 min. The maximum observed mixing ratio of $CH_4$ in the gas phase concentration was 77%, which at 150 m of depth and thermal condition of the lake, corresponds to 3.5 mmol $L^{-1}$. Deeper down, dissolved $CH_4$ concentrations were too large for the methane absorption spectrum to be correctly retrieved. Results are in good agreement with discrete in situ measurements conducted with the commercial HydroC® sensor. This fast profiling feature is highly profitable to study the transport, production and consumption of $CH_4$ and other dissolved gases in aquatic systems. While the sensor is well adapted for investigating most of environments with concentration of $CH_4$ up to few mmol $L^{-1}$, in the future the spectrometer could be replaced with a less sensitive analytical technique possibly including simultaneous detection of dissolved $CO_2$ and total dissolved gas pressure, for exploring settings with very high concentrations of $CH_4$ such as the bottom waters of lake Kivu.

## 1 Introduction

Methane ($CH_4$) is the second most important greenhouse gas contributing to the anthropogenic radiative forcing of the atmosphere and its atmospheric content raised by 2.5 times since the industrial age. During the last decades, significant efforts have been made to better estimate methane contributions of natural and anthropogenic sources to the global atmospheric budget (Kirschke et al., 2013; Saunois et al., 2019). The development of more advanced techniques allowed the recognition of a larger

number of sources which, coupled with the improvements in the modelling, led to continuous rectifications of this budget (Hamdan and Wickland, 2016). In the last three decades, natural sources contribute for ~35-50% of the total global methane emissions, and freshwater constitutes one of the largest fluxes after natural wetland and together with geological sources (including seafloor). This highlights the importance and urgency for a better inventory of the sources of $CH_4$, and to reduce

the uncertainties of the contributions of aquatic systems (lakes, rivers, estuaries, coastal seas and open ocean) (Ciais et al., 2013). Fast response instruments for in situ dissolved gas measurements and dynamic profiling can provide the data for a better understanding of the undergoing processes of production, transport, and transformation.

     In this work, a fast response prototype instrument was deployed for the first time at lake Kivu, located in East Africa at the border between Rwanda and the Democratic Republic of the Congo. The meromictic character of this lake, defined by a

strong stratification of the water, makes deep water strongly decoupled from surface layer because of their difference in density and composition (Schmid and Wüest, 2012). The upper tens of meters (ranging from 65 to 25 m depending on seasons) correspond to the oxic zone, while deeper waters are anoxic and contain large amount of dissolved carbon dioxide ($CO_2$) and $CH_4$, with the strongest chemocline situated at 250 m of depth (Schmid et al., 2005). Since 1935, several measurement campaigns have been carried out, aiming at quantifying the amount of dissolved $CH_4$ and $CO_2$ present in the lake (e.g. Degens et al., 1973; Pasche et

al., 2011; Schmitz and Kufferath, 1955; Tassi et al., 2009; Tietze et al., 1980). On the one hand, the presence of those gases constitutes a risk of catastrophic event such as a gas eruption, which in the past already occurred in other gas-rich lakes (e.g. in 1984 at lake Monoun and in 1986 at lake Nyos in Cameroon (Kling et al., 1987; Kusakabe, 2017; Sigurdsson et al., 1987)). On the other hand, dissolved $CH_4$ represents a potentially important energy resource. Methane extraction would allow to compensate further accumulation of gas at the bottom of the lake and therefore preventing the possibility of a gas eruption. From this field

campaign, the maximum total dissolved gas pressure (TDGP) was estimated to be 50±7 % of the hydrostatic pressure at 320 m of depth (Bärenbold et al., 2019; Schmid et al., 2019). Meanwhile, extraction has to be performed without destabilizing the stratification of the lake or altering its ecosystem. Regarding the stability of the lake, in 2005 Schmid and co-workers raised the possibility that dissolved $CH_4$ in the lake was increasing with a rate of ~0.5 % per year, with consistent repercussion on the safety of the surrounding population (Schmid et al., 2005). However, from the work of Pasche et al. (2011) as well as the results from

this recent field campaign, the hypothesis of a fast increase is today excluded, and the temporal variability appears to be slower than previously expected (Bärenbold et al., 2019; Boehrer et al., 2019; Schmid et al., 2019). In the future, regular monitoring of the lake is required to estimate the $CH_4$ and $CO_2$ budgets as well as their temporal variability, using reliable, fast and easy to use techniques. For a more precise estimation of the dissolved gas concentration, inter-comparison between different sensors and methods is required, as conducted and presented in this work and in the even more comprehensive results from the entire inter-

comparison campaign (Bärenbold et al., 2019; Boehrer et al., 2019; Schmid et al., 2019). A fast response sensor as the one proposed here could also be highly profitable for estimating methane fluxes from the water surface and their spatial and seasonal variabilities.

     In this work we report a successful deployment of the Sub-Ocean sensor in a very different setting, highlighting the reliability and adaptability of the technique to different aquatic environments. Advantages and drawbacks of the technique are highlighted

in the discussion section in comparison with other methods deployed during the same campaign: water sampling followed by laboratory gas chromatography analysis (Boehrer et al., 2019) and on-line water pumping followed by on-site mass spectrometry analysis (Brennwald et al., 2016). These results are not reported here as they focused on the concentrations in the deep waters (Bärenbold et al., 2019; Schmid et al., 2019).

## 2 Materials and Methods

### 2.1 The Sub-Ocean Instrument

The optical instrument used in this study is based on the OFCEAS technique (optical feedback cavity enhanced absorption spectroscopy) (Morville et al., 2003, 2014) developed for trace gas sensing. The dissolved air from the extraction unit (Figure 1) is continuously pumped toward the optical cavity of the spectrometer. The internal volume of the cell is less than 20 cm$^3$ and provides sample residence times < 30 sec for optimal running conditions (compromise between the cell pressure and the

total gas flow).

Extraction of dissolved gases from water is performed using a silicon PolyDiMethylSiloxane (PDMS) membrane. The extraction technique does not rely on gas equilibration across the membrane but, in order to achieve fast response, the dry side of the membrane is maintained at low pressure while continuously flushing it with dry zero air (Triest et al., 2017). The pressure at the dry side controls the total flow of dry and wet air through the membrane, and the system is designed to keep this pressure

constant. While the spectrometer operates at about 20 mbar, the pressure at the dry side of the membrane is maintained at about 30 mbar.

A full description of the in situ membrane inlet laser spectrometer instrument (Sub-Ocean), together with the experimental setup used for laboratory calibrations can be found in Grilli et al., 2018. In order to adapt the instrument to the high concentrations of dissolved CH$_4$ expected in lake Kivu, the absorption spectrum of the optical spectrometer was set away from

the strong CH$_4$ rotational-vibrational transitions, more precisely at 2238.5 nm, where concentrations inside the optical cavity may reach up to 1.5 - 2 % of CH$_4$ in air before optical saturation (equivalent to an absorption $10^{-5}$ - $10^{-6}$ cm$^{-1}$). Above this absorption, the transmission signal at the maximum of the peak of absorption becomes too weak and the optical feedback to the laser, required by the optical method, is no longer strong enough to lock the laser frequency for a period of time close to the cavity free spectral range. This leads to narrower cavity modes and to a failure in correctly retrieving the absorption features.

A stainless-steel membrane block (MB) was equipped with two 10 μm thick PDMS membranes of 56 mm diameter mounted face-to-face. The thin-film membranes were mounted on porous bronze frits of 3 mm of thickness (Poral, grade 20), providing mechanical strength for the membrane under high hydrostatic pressure. A schematic of the membrane block can be found in the supplementary information of Grilli et al., 2018. For this campaign, in order to increase the dynamic range of the measurements, one of the two membranes was replaced with a gas-tight Teflon film. This increased the dilution factor by

decreasing the flow of the permeating gas with respect to water vapor and carrier gas flow, but degrades the precision of the measurements due to the low dry gas flow through the membrane. A picture of the instrument and the assembly taken during

the campaign is shown in Figure 2. The main (central) pressure tube (140-cm long, 28-cm diameter) is mounted on a metal frame. The membrane block at the bottom is connected with a submersible water pump (Sea-Bird Electronics, SBE 5T) providing a flow of 0.8 L min$^{-1}$ along the membrane. A 1 L carrier gas (CG) tank containing dry zero air at a pressure between 2 and 40 bar, depending on the suitable autonomy, is attached on the frame and connected to the instrument via a 1/8" stainless-steel tube. A subsea battery (Seacell, STR) was mounted on the metal frame, providing up to 12 hours of continuous operation. An independent CTD (Sea & Sun Marine Tech, CTD-60) was also attached to the frame for depth, temperature, conductivity and dissolved oxygen measurements. For an operation where the instrument is powered through an electromechanical cable the autonomy will be limited by the storage of the dry gas inside the instrument housing. For fast response measurements, at maximum carrier gas flow of 6 ml min$^{-1}$ is required corresponding to an autonomy of 24h, whereas without the use of carrier gas the autonomy will stretch to 90 days since most of the gas flow will be composed of water vapor that is trapped before the vacuum pump by the silica gel dryer (however, the long-term deployment may be limited by the capability of the silica gel).

The embedded spectrometer is continuously measuring the gas composition at 10 Hz, while the response time of the sensor during the campaign, expressed as $\tau_{90}$, was ~10 sec. At a lowering speed of ~6 m min$^{-1}$, the vertical resolution is 1 m. From the composition of the dissolved gas the instrument can indirectly estimate the amount of $N_2$. This requires to know: TDGP, $pCO_2$ and $pO_2$ which were not measured by the Sub-Ocean probe and rely on other sensors. The partial pressure of $N_2$ can then be estimated as $pN_2 = TDGP - pCH_4 - pCO_2 - pO_2$.

## 2.2 The HydroC-CH$_4$ commercial instrument

In situ discrete measurements of dissolved $CH_4$ at five different depths along the upper 150 m of the water column were performed using a commercial equilibrium-based underwater sensor, the Contros HydroC® HP sensor. The dissolved gas diffuses from the liquid through a thin film composite membrane into an internal gas cell. Therein, the total dissolved gas pressure and the partial pressure of $CH_4$ gas are measured by a pressure sensor and a non-dispersive infrared spectrometer, respectively. The HydroC® $CH_4$ HP sensor is similar to the HydroC® $CO_2$ sensor presented in Fietzek et al., 2014, except for the absence of an internal zeroing system and a $CH_4$-specific fixed narrow-band spectral filter from 3.3-3.4 µm. The sensor was calibrated in October 2012 and November 2015 by the manufacturer. The calibrations were made using a specially designed pressure chamber with fresh water brought to pressure using compressed target gas. Three standard gas mixtures of $CO_2$, $CH_4$ and $N_2$ (100 % pressure $N_2$; 50 % pressure $CH_4$ and 50 % pressure $CO_2$; 100 % pressure $CH_4$) were used to equilibrate the water volume along a gas pressure gradient (5-6 points) from 1 up to 30 bars and partial pressures of $CH_4$ from 0.5 to 18 bars. The calibration results showed the absence of a significant drift of the sensor (< 3 % within the lake Kivu gas concentration range) between the October 2012 and November 2015 calibrations. Also, several $CH_4$ profiles were carried out in lake Kivu from 2016 to 2018 using the HydroC® $CH_4$ HP sensor and the repeatability of the observed $CH_4$ partial pressures was 3.8 % (2σ) below the main density gradient. However, the calibration curve as a function of the methane concentration

was determined by using three points (0, 50 and 100% CH$_4$), and because of the nonlinear behavior of the detection system, a systematic error could be present, but it should not exceed 10 % (manufacturer personal comm.).

The HydroC® CH$_4$ HP system was mounted on a SeaBird 19plus V2 SeaCAT CTD profiler equipped with a SBE 43 Dissolved Oxygen sensor and a SBE 18 pH sensor. Calibrations of the SeaBird sensors were performed following manufacturer instructions. Water circulation in front of the membrane was provided by a SeaBird 5T pump, ensuring a continuous and homogeneous water flow at the membrane. A zero calibration of the system was made daily before each deployment using surface waters. The sampling rate was 1 Hz. The steady-state of the sensor was generally reached within 40 minutes and real-time data communication using an electromechanical cable allowed to adjust the waiting time at each depth accordingly. In all cases, the waiting time for each depth never exceeded 1 hour. The retained partial pressure of CH$_4$ is the average for the last 5 min of the equilibration curve.

## 2.3 Calculation of dissolved CH$_4$

Both the Sub-Ocean and the HydroC® HP sensors measure CH$_4$ in the gas phase, and raw data are expressed as the concentration of CH$_4$ with respect to the total amount of dry gas permeating the membrane. For the Sub-Ocean system, the concentration of CH$_4$ in the dry gas downstream from the membrane [CH$_4$]'$_g$ can be expressed with respect to the expected concentration of the gas in the headspace which would be in equilibrium with the water sample, [CH$_4$]$_g$. In eq. 1, Pr are the membrane permeability coefficients for CH$_4$ and X (N$_2$, O$_2$ and CO$_2$) reported in Robb (1968), but corrected for their temperature and salinity dependency.

$$[CH_4]'_g = \frac{Pr_{CH_4} \cdot [CH_4]_g}{\sum Pr_X \cdot [X]_g} \quad , \tag{1}$$

Concentrations, [CH$_4$], [X] are expressed as mixing ratios. Measuring the concentration of water vapor [H$_2$O]$_g$ is required in order to retrieve the dissolved CH$_4$ concentration, [CH$_4$]$_{diss}$, since water vapor flow will cause dilution of the measured dry gas mixture (as well as the carrier gas flow). This measurement is performed by the OFCEAS spectrometer embedded in the Sub-Ocean probe, simultaneously with the CH$_4$ measurement. Precision on the water vapor concentration was ± 0.6 % (2σ). [CH$_4$]$_{diss}$ is then calculated from the following equation:

$$[CH_4]'_{diss} = \frac{[CH_4]'_g \times f_t}{f_t - f_{CG} - (f_t \times [H_2O]_g)} \times \frac{1}{m_{eff}} , \tag{2}$$

where [CH$_4$]'$_g$ represents the methane mixing ratio measured by the optical spectrometer, $f_t$ and $f_{CG}$ are the total- and carrier-gas flow (ml min$^{-1}$), respectively, and [H$_2$O]$_g$ corresponds to the mixing ratio of water permeating through the membrane. The

denominator term $(f_t - f_{CG} - (f_t \times [H_2O]_g))$ corresponds to the dry flow permeating the membrane. $m_{eff}$ represents the enrichment

factor due to the membrane and corresponds to the quantity $\frac{Pr_{CH_4}}{\sum Pr_x \cdot [X]_g}$ in eq. 1. Its dependency with temperature and salinity is

calculated by running calibrations under various conditions (Grilli et al., 2018). From our calibration, a $m_{eff}$ of 2.84 ± 0.11 for

fresh water at 25°C and 1.2 bar was calculated. This is in agreement with an expected value of 2.76 calculated from the

permeation coefficients reported by Robb (1968).

As reported in eq. 1 above, this technique requires to know the main composition of the dissolved gas, in order to account

for the different permeation coefficients of the species through the PDMS membrane. This does not represent a problem for

most of the ocean and lake settings, where the gas mixture is mainly composed of nitrogen and oxygen, but it requires a more

complex analysis for a setting such as lake Kivu. For the data analysis we assumed a bulk gas mainly composed of $N_2$, $O_2$,

$CO_2$ and $CH_4$. $H_2S$ is only present in bottom water and in lower amount with respect to $CO_2$ and $CH_4$, and was therefore

neglected here. Oxygen concentrations were calculated from the CTD measurements and converted into partial pressures using

equation 19 from Sander 2015 (using $H^{cp}$ of $1.25 \times 10^{-5}$ mol m$^{-3}$ Pa$^{-1}$ and $d\ln(H^{cp})/d(1/T)$ of 1500 K).

    As mentioned above, concentrations reported so far are expressed in mixing ratio with respect to the total dissolved gas

pressure TDGP. Therefore, by knowing the TDGP, a value of partial pressure, $pCH_4$, can be retrieved which is then converted

into dissolved methane concentrations, $C_{CH4}$, expressed in mol per liter of water. This conversion is performed by considering

the solubility of the gas in water under given physical conditions as well as its fugacity. The procedure has been previously

described in a scientific report (Schmid et al., 2019). $C_{CH4}$ is related to the $pCH_4$ through the following equation:

$$C_{CH4} = K(T, S, P)\, pCH_4\, \varphi_{CH4}(T, P) \;,\tag{3}$$

where $\varphi_{CH4}$ is the fugacity coefficient, i.e. the ratio between the fugacity of a gas and its partial pressure, which is a function

of temperature $T$, pressure $P$ and gas composition, and $K$ is the solubility coefficient, i.e. the ratio between the dissolved

concentration of a gas and its fugacity. The solubility coefficient K (mol L$^{-1}$ atm$^{-1}$) of $CH_4$ as a function of temperature T (K)

and salinity S (g/kg) is calculated using the following equation:

$$ln(K) = A_1 + A_2(100/T) + A_3 \ln(T/100) + S[B_1 + B_2(T/100) + B_3(T/100)^2] \;,\tag{4}$$

The parameters in eq. 4 are from Wiesenburg and Guinasso, 1979.

The solubility coefficients need to be corrected for the local pressure P (bar) at the sampling depth (sum of hydrostatic pressure

plus atmospheric pressure), using the following equation (Weiss, 1974):

$$K(P) = K e^{\left[\frac{(1-P)\, v_{CH4}}{RT}\right]} \;,\tag{5}$$

where R = 83.1446 cm$^3$ bar K$^{-1}$ mol$^{-1}$ is the gas constant, and $v_{CH4}$ is the partial molar volume (cm$^3$ mol$^{-1}$) of $CH_4$ calculated

from Rettich et al., 1981.

The fugacity coefficients were calculated using the methods described in Ziabakhsh-Ganji and Kooi, 2012. A Maple script was provided by Z. Ziabakhsh-Ganji, which was transcribed to Matlab code by M. Schmid (Schmid et al., 2019). The script calculates, among other things, the fugacity coefficients for $CO_2$ and $CH_4$, including the interactions between both gases.

## 2.4 The lake and the field campaign

Lake Kivu [2.50°S - 1.59°S ; 29.37°E - 28.83°E] located at 1460 m above sea level, has a surface of 2 700 km$^2$ (of which 2385 km$^2$ represents the water covering) and a maximum depth of ~485 m. The measurement campaign took place from 9th to 13th March 2018 at ~6 km from Goma and ~5 km from Gisenyi/Rubavu at the Northern shore of the lake (1.74087°S - 29.22602°E) and nearby a permanent platform with water depth of 410 m. During the campaign other types of measurements of dissolved methane and carbon dioxide were performed. The research team from Eawag (Switzerland) analyzed pumped water on the platform using a field mass spectrometer instrument (Brennwald et al., 2016), while a second team from UFZ (Germany) sampled water from a boat and measured the samples by head-space equilibration and gas chromatography (GC) analysis at the Lake Kivu Monitoring Program (LKMP) laboratory in Rubavu (Boehrer et al., 2019). The Sub-Ocean sensor was deployed from a research boat during three days of the campaign: 10[th], 12[th] and 13[th] of March, with a total of eight continuous profiles. Measurements with the commercial HydroC® HP sensor were conducted during the campaign and on May 8th -11th at the same location as the Sub-Ocean measurements and over specific discrete depths.

## 3 Results and Discussions

In Figure 4 an example of a consecutive downward and upward profile of dissolved $CH_4$ measured by the Sub-Ocean sensor is reported. $CH_4$ concentrations are expressed as mixing ratio with respect to the total dissolved gas. The sensor was lowered at a speed of ~6 m/min, reaching 100 m depth in only 18 min. The response time of the sensor during the campaign expressed as $\tau_{90}$ was ~10 sec, which corresponds to a vertical resolution of 1 m. On the right-hand side, dissolved $CH_4$ is plotted against depth, showing the reproducibility of the sensor during descent and ascent.

A total of eight continuous profiles (downward and upward) were obtained with the Sub-ocean instrument during the campaign. They are reported in Figure 5 together with dissolved $CO_2$, CTD data (temperature, conductivity and dissolved oxygen) and total dissolved gas pressure (TDGP). For the measurement of $CH_4$ only one of the eight profiles reached 150 m, while the others are shallower, only covering the upper 100 m of depth. The accuracy of the measurement was estimated at 80 m depth, where water mass is well stratified. At this depth, an average concentration of 35.5 ± 7.8%, corresponding to 508.3 ± 112 mbar of partial pressure and 0.71 ± 0.16 mmol L$^{-1}$ of $CH_4$ was calculated, leading to a repeatability of ± 22% (2σ). This relatively large standard deviation can be explained by the large uncertainty in determining the total flow of dry gas permeating the membrane. The value is in agreement with previously observed performances, where an error propagation of ±12% (2σ) was calculated using two semipermeable membranes (Grilli et al., 2018). The use of only one membrane allowed to further

increase the dynamic range of the sensor by diluting the dry gas permeating the membrane. However, in this condition, a dry

gas flow of only ~0.065 cm$^3$ STP/min is delivered by the extraction system. The large uncertainty on this dry flow measurement directly affects the accuracy on the retrieved concentration. The uncertainty represented by the grey lines in Figure 6 represents the measured variability over the eight vertical profiles from 0 to 80 m, and was fixed to ± 22% at larger depths. The $CO_2$ data are from Schmid et al. 2005 and are calculated from alkalinity and pH measurements. TDGP are discrete measurements at seven different depths measured with the HydroC® HP sensor which have been interpolated to match the depth resolution of

the Sub-ocean data. Nitrogen ($N_2$) mixing ratio was retrieved assuming that the main gas is composed by $N_2$, $CO_2$, $CH_4$, and $O_2$ ($pN_2$ = TDGP – $pCH_4$ - $pCO_2$ - $pO_2$).

The molar concentrations as a function of depth for the average continuous profile recorded by the Sub-Ocean sensor and for the discrete measurements obtained with the HydroC® HP sensor are reported in Figure 6. A good agreement between the two independent measurements is observed. The measurements were obtained during the same field campaign at the measurement

site location near Goma (the two vessels were a few hundred meters away from each other). However, the measurements were not performed simultaneously. In the graph, results from previous campaigns are also reported. Data from the University of Liege obtained during a long-term monitoring of the lake are reported in orange. Data were collected from June 2011 to August 2014 at different periods of the year (both dry and rainy seasons) and at different locations (northern and south basin) (Roland et al., 2017, 2018). The large variability of these measurements is reported by the orange lines (Figure 6) defining the 3σ

distribution. Data from the works of Pasche et al. 2011 and Schmid et al. 2005 are also reported in green and blue, respectively. The measurements from ULiege and Pasche 2011 were obtained by sampling the water using Niskin bottles and analyzing the dissolved gas in the laboratory by head-space technique followed by GC analysis. The others (this work and Schmid 2005) are from in situ measurements. From the data, one can see that below 80 m depth, where the TDGP becomes larger than atmospheric pressure (1.4 bar at 80 m, Figure 5), a problem due to degassing of the sample collected on the Niskin bottles was

observed, leading to an under-estimation of the dissolved $CH_4$. Data from Schmid 2005, which are from a commercial Capsum Met sensor (Franatech) and data from the Contros sensor are a bit lower than the measurements with the Sub-Ocean probe at higher concentrations (and depths), but they still lie within the measurement uncertainties. During the campaign the HydroC® HP sensor also showed a good agreement with the other discrete techniques (on site mass spectroscopy and discrete sampling followed by GC analysis) between 150 and 250 m, while at greater depths, the HydroC® HP values were lower by ~12%

(Schmid et al., 2019). This may be due to a problem of calibration of the sensor at high hydrostatic pressures, but it requires further investigations to be confirmed. Regarding the Capsum Met sensor, no information about the calibration of the sensor were found, therefore no further discussion can be carried out.

Surface measurements performed by the Sub-Ocean instrument lead to average concentrations of 0.59 ± 0.03 μmol L$^{-1}$ and 0.72 ± 0.14 μmol L$^{-1}$ over the upper 10 and 30 m, respectively. Those values sit at the higher edge of the observed average

seasonal concentrations, which span from 0.008 to 11 μmol L$^{-1}$ (Roland et al., 2017, 2018, and more recent unpublished data from the same autors). Despite the large seasonal and spatial variability, our results are in good agreement with the one from

Pasche et al 2011 which were obtained at a similar time of the year but at different locations (May 2006 and 2007 in Kibuye, Gisenyi and Ishungu). A stronger similarity can be found with the dataset from the same location (Gisenyi 2007) in the northern basin. CTD measurements (temperature, conductivity and dissolved oxygen, Sea & Sun Marine Tech, CTD-90M) performed a few months prior to the campaign at the research platform (Figure 7) confirmed a typical behavior of the lake stratigraphy while going from a dry into a rainy season (Roland et al., 2017) and justified therefore the high concentrations measured in this work. The lake was mixed down to at least 50 m depth during the previous dry season, and started to stratify in mid-December, leading to a 25-m depth seasonal thermocline. Below the thermocline, $O_2$ was rapidly consumed by mineralization of organic matter and oxidation of reduced compounds (e.g. methane, ammonium) diffusing upward. By the end of February, $O_2$ supplied at these depths during the previous dry season was completely vanished. Then, on the first-half of March, a mixing event occurred down to about 35 m depth, favoring the mixing between anoxic water (35-25 m depth), enriched in dissolved $CH_4$, and surface water. From the top 10 m layer temperature profiles reported in Figure 7 one can see that by March 22$^{nd}$ the temperature slope disappeared, supporting the occurrence of the water mixing. Unfortunately, the reasons for this mixing event are still unknown. Meteorological records from December 2017 to March 2018 do not indicate neither high wind speed, low temperature, or low relative humidity events that could support our observations. Comparing the second-half of February to the first-half of March, average temperatures decreased by 1°C (from 21.2 to 22.2°C) and average precipitations increased by a factor of two, with peaks up to 7.6 mm of rainfall on March 6$^{th}$. As reported by Rooney et al., 2018, rain may have a cooling effect on the lake surface by lowering the near-surface air temperature and inducing a convective mixing of the lake surface layer. Finally, $CH_4$ concentration in the surface layer may depend on the biogeochemical processes such as for instance the methanotrophy. Further investigations are therefore required to better understand the dynamic of the surface layer of the lake at this period of the year.

This type of fast response sensors could be used to better investigate the fluxes of $CH_4$ (or other greenhouse gases) from lakes, oceans, rivers and other water reservoirs. In this campaign, only a specific location at 5 km from the coast with 410 m of water depth was investigated. The amount of $CH_4$ at the surface may strongly depend on the water depth, i.e. on the distance of the sediment to the surface, as well as to the horizontal distance from the shore and littoral sediments (DelSontro et al., 2018b). A fast sensor would allow to follow the spatial distribution of dissolved gases at the surface layer, as well as its variability over the seasons. This would help to better constraint the greenhouse gas emissions in the face of global change (DelSontro et al., 2018a). Beside the advantages of the Sub-Ocean probe to provide in situ, continuous and fast measurements, some drawbacks of the technique can be identified: **i)** the instrument was designed for measuring background concentrations in the oceans (sub-nmol L$^{-1}$) while lake Kivu reaches ~18 mmol L$^{-1}$ in bottom waters, thus with eight orders of magnitude difference. Despite the efforts to make the sensor less sensitive, the Sub-Ocean could not measure below 150 m depth, corresponding to a maximum measurable concentration of 3.5 mmol L$^{-1}$, where absorption becomes too strong for the optical spectrometer at the selected laser frequency. **ii)** In such environment, a good knowledge of the total dissolved gas pressure and of the concentration of dissolved $CO_2$ are required for correctly determine the concentration of $CH_4$. Those parameters were measured during the field campaign, but they are not currently integrated in the sensor. This could be performed in the future

by detecting simultaneously $CO_2$ and $CH_4$ using the same gas analyzer and by integrating the TDGP measurement or deploying the sensor with an independent TDGP device. It should be noticed that TDGP sensors have response times of a few minutes (e.g. $\tau_{63}$ = 2 min for the Mini-TDGP from Pro-Oceanus) which could be a limiting factor with respect to the faster response time of the Sub-Ocean sensor. **iii)** Because a small dry gas flow through the membrane was required (in order to increase the dilution factor), the precision of the measurement was degraded by a factor of two with respect to previous deployments, leading to a ±22% precision. By using a less sensitive gas analyzer, the above drawbacks could be avoided, or at least minimized, making the technique fully suitable for monitoring meromictic lakes with a large range of dissolved $CH_4$ concentrations.

It should be noticed that different lakes have different dissolved $CH_4$ concentration ranges. Lake Kivu represents a very high range (with ~18 mmol $L^{-1}$ at the bottom) while for instance lake Pavin in France or lake Vollert-Sued in Germany both reach concentrations up to few mmol $L^{-1}$ (Horn et al., 2017; Lopes et al., 2011) making the Sub-Ocean probe in its current status well suitable for acquiring continuous full vertical profiles at those sites.

## 4 Conclusions

The comparison between different types of measurements confirms the reliability of the fast response membrane extraction system of the Sub-Ocean sensor under more extreme conditions (in terms of dissolved gas content) than ocean settings. Lake Kivu is particularly challenging because of the high amount of dissolved $CH_4$ and $CO_2$ as well as their large variability. The gas composition strongly varies across the oxic-anoxic boundary and further down across the different chemoclines, going from a background composed by $N_2$ and $O_2$, to another one which sees $CH_4$ and $CO_2$ as the main dissolved gases. The Sub-Ocean sensor allowed fast vertical profiles of $CH_4$ which are in good agreement with the discrete in situ measurements made with the commercial HydroC® HP sensor at five different depths. During the campaign the HydroC® HP sensor also showed good agreement with the other discrete techniques (on site mass spectroscopy and discrete sampling followed by GC analysis) between 150 and 250 m. At 80 m of depth, where no spatial variability of the dissolved gas is expected, an accuracy of ±22% (2σ) was estimated for the Sub-Ocean probe by comparing the eight independent profiles at this depth. The maximum measurable concentration of dissolved $CH_4$ was 3.5 mmol $L^{-1}$ at 24°C, 150 m of depth, and TDGP of 2.62 bar, which corresponds to a mixing ratio of 77% with respect to the total dissolved gas.

An average concentration of $0.59 \pm 0.03$ µmol $L^{-1}$ of $CH_4$ was found in the 10-m surface layer, which sits at the higher edge of the observed average seasonal concentrations of the lake. The variability of the physical parameters during a period of three months prior the campaign suggests a mixing event of the top 35 m, which can explain the high values measured at the surface. The causes of this mixing event are however not clear and further investigations will be required to better understand the behavior of the lake while going from the dry into the rainy season.

Such a campaign highlights the advantages of using the Sub-Ocean technology for measuring the dissolved gas content in meromictic lake settings. The technology allows in situ, continuous and fast profiling, important for a long-term monitoring of water resources. The in situ deployment prevents any possible contamination and artefact of the measurement due to water and/or gas sampling and subsequent laboratory analyses. The fast response of the instrument would allow to complete a full vertical profile over 470 m of depth with 1 m resolution within ~1h 20min, while current techniques of in situ discrete measurements would take more than 1h per measured depth. The measurement by this technique has now been proven over a very large dynamic range of seven orders of magnitude, spanning from sub-nmol $L^{-1}$ in open ocean waters to mmol $L^{-1}$ concentrations of dissolved $CH_4$ and in a context of very different dissolved gas composition and TDGP. The instrument is therefore well suitable for fast profiling on different water reservoirs, and could be further adapted to the entire vertical column of lake Kivu by using a less sensitive gas analyzer.

**Author Contributions:** RG, JT and JC are the inventors of the Sub-Ocean instrument. JC, AM and AU initiated the collaboration leading to the field campaign. AM and AU organized the field campaign and took care of the project administration. RG optimized the instrument for the measurements at lake Kivu and ran the laboratory calibrations. RG prepared the instrument for the field and was in charge of the field campaign with the Sub-Ocean instrument. FD handled the measurements with the HydroC® HP sensor and its data analysis. JT contributed to the analysis of the HydroC® HP sensor and TDGP data. RG analyzed the Sub-Ocean data. All authors contributed to the manuscript.

**Funding:** This research was funded by the European Community's Seventh Framework Programme ERC-2015-PoC, grant number 713619 (ERC OCEAN-IDs), the European Community's Seventh Framework Programme ERC-2011-AdG, grant number 291062 (ERC ICE&LASERS) and the National Research Funding, ANR, through the program ANR-18-CE04-0003-01.

**Acknowledgments:** The research leading to these results has received funding from the European Community's Seventh Framework Programme ERC-2015-PoC under grant agreement no. 713619 (ERC OCEAN-IDs). The work was made possible thanks to pioneering investigations conducted under the European Community's Seventh Framework Programme ERC-2011-AdG under grant agreement no. 291062 (ERC ICE&LASERS) and with support from SATT Linksium of Grenoble, France, and of the Service Partenariat &Valorisation (SPV) of the CNRS. The authors would like to thank Martin Schmid for his profitable help and discussions regarding the conversion from partial pressure into concentration units. Fabian Bärenbold, Bertram Boehrer, Wolf von Tümpling, Placid Nkusi, Maximilian Schmidt, Eric Mudakikwa, Irénée Nizere, Gaeta Sakindi, the captain of the vessel for their help during the field campaign and their helpful discussions. Thanks to the entire Lake Kivu Monitoring Program for the organization of the field campaign. We thank Wim Thiery and Nicole van Lipzig for providing the meteorological data. We also thanks Alberto Borges and Fleur Roland and Martin Schmid for the discussions on the surface

water mixing. Véronique Gosselain (University of Louvain, Belgium) is thanked for her networking to put the Lake Kivu Monitoring Program and scientists from IGE, Grenoble, France, in connection.

**Conflicts of Interest:** The authors declare no conflict of interest.

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

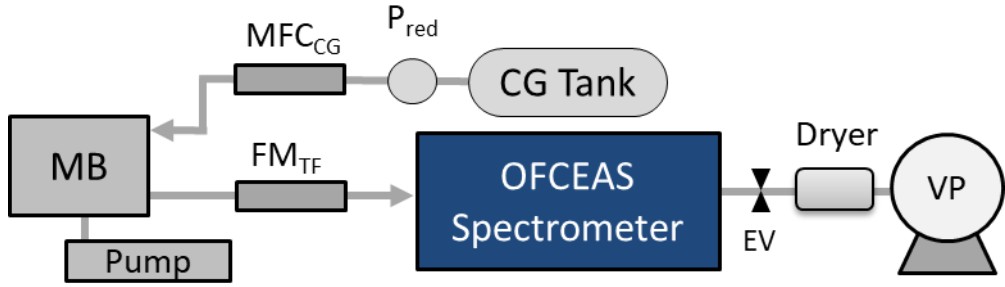


**Figure 1**. A schematic of the Sub-Ocean sensor. MB is the membrane block where the gas extraction occurs. Water circulates at the membrane using a submersible pump. The carrier gas (CG) flow is controlled by a mass flow controller (MFC$_{CG}$) and the flowmeter FM$_{TF}$ is used for monitoring the total gas flow. The low pressure on the optical spectrometer is provided by a vacuum pump (VP) and an electronic valve (EV). P$_{red}$ is a pressure reducer. A silica gel dryer is placed before the VP for trapping water vapor.


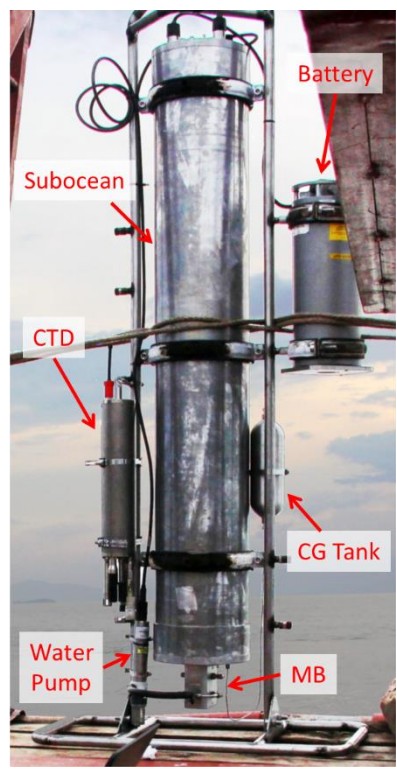

**Figure 2**. A picture of the Sub-Ocean instrument and the full assembly. The sensor is mounted on a metal frame. The main tube at the center is 150-cm long and 28-cm diameter. The membrane block (MB) at its bottom is connected to the water pump to ensure a constant flow of water against the membrane. The carrier gas (CG) tank is attached to the metal frame and connected with a 1/8" stainless-steel tube at the instrument. An STR battery pack and a CTD sensor were also attached to the metal structure. The total weight of the assembly is 120 kg with about -50 kg of buoyancy.


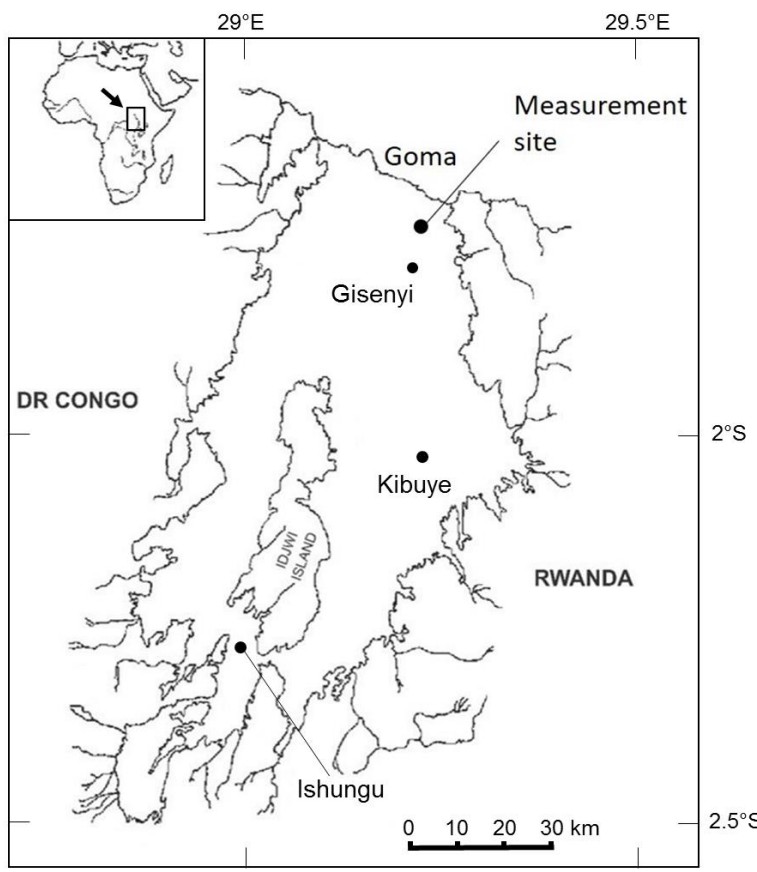

**Figure 3**. Map of lake Kivu showing the location of the measurement site. Locations of previous campaigns mentioned in the discussion part are also reported (named Gisenyi, Kibuye and Ishungu).


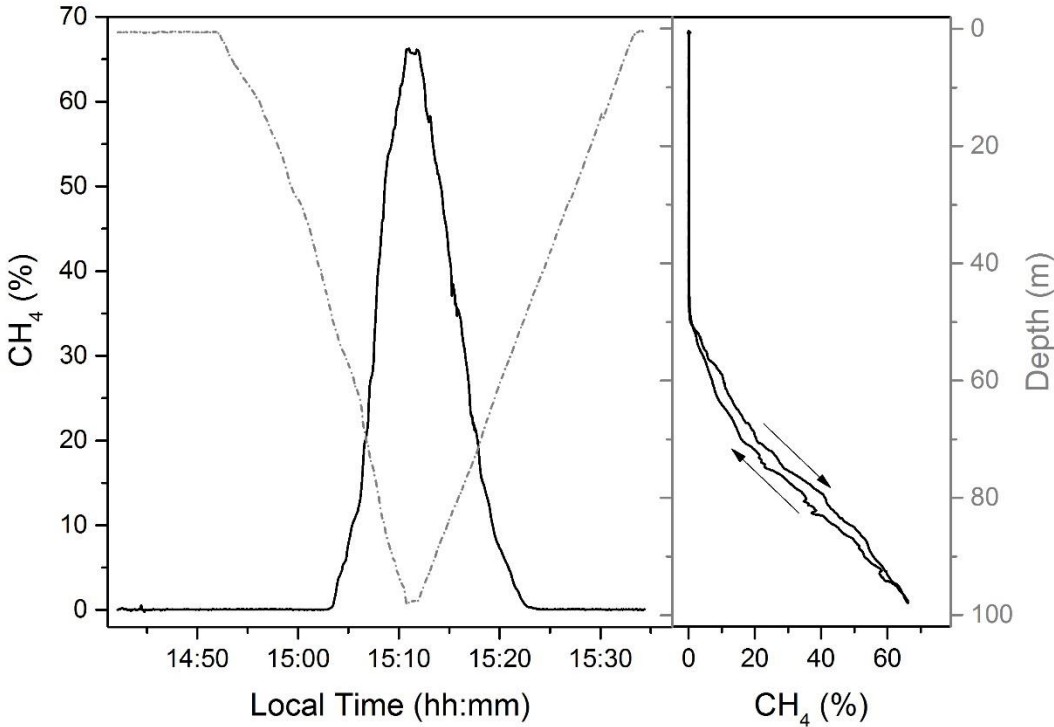

**Figure 4**. One of the methane continuous profiles recorded by the Sub-Ocean on 10[th] March 2018. The concentration is expressed as a percentage of CH$_4$ with respect to the total dissolved gas. The 100 m downward and upward profile was recorded in 42 min. On the right panel the two profiles are superposed, highlighting the reproducibility of the measurement between descent and ascent.

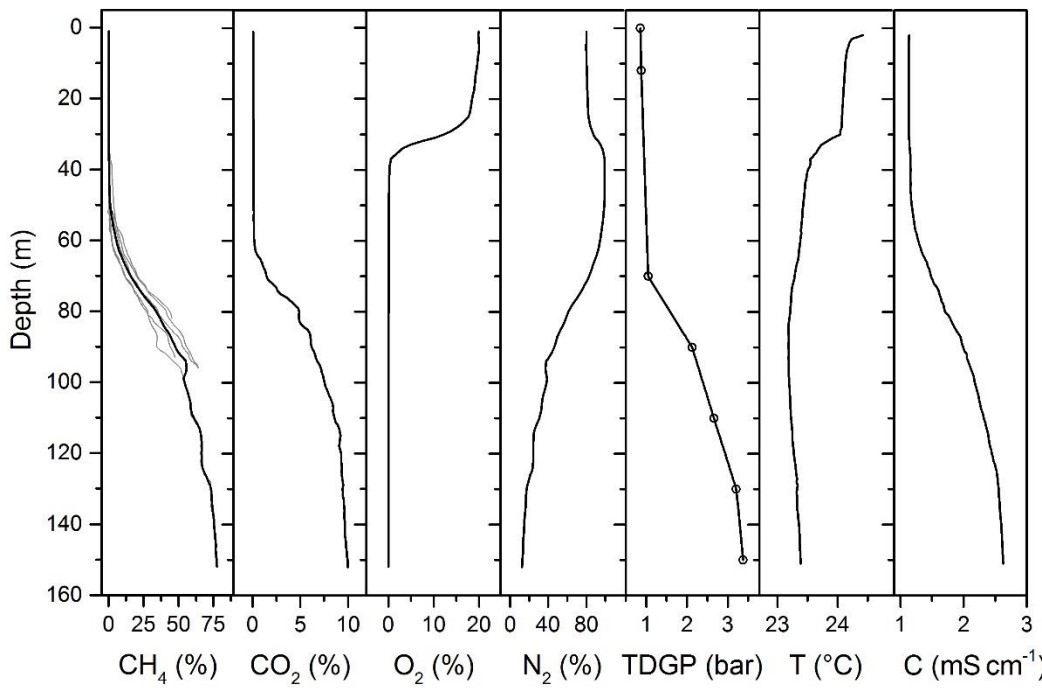

**Figure 5**. Mixing ratios of individual gas species in the dissolved gas mixture and total dissolved gas pressure. Grey $CH_4$ lines represent the eight profiles recorded by the Sub-Ocean instrument during the campaign, while the black line is the averaged value. $CO_2$ data are from (Schmid et al., 2005), $O_2$, temperature and electrical conductivity are from CTD data during the campaign, and $N_2$ is a concentration profile deduced from the other measurements (TDGP – $pCH_4$ - $pCO_2$ - $pO_2$). The total dissolve gas pressure, TDGP, was measured using the Contros HydroC® HP sensor (open circles), the black line is an interpolation of the data. .

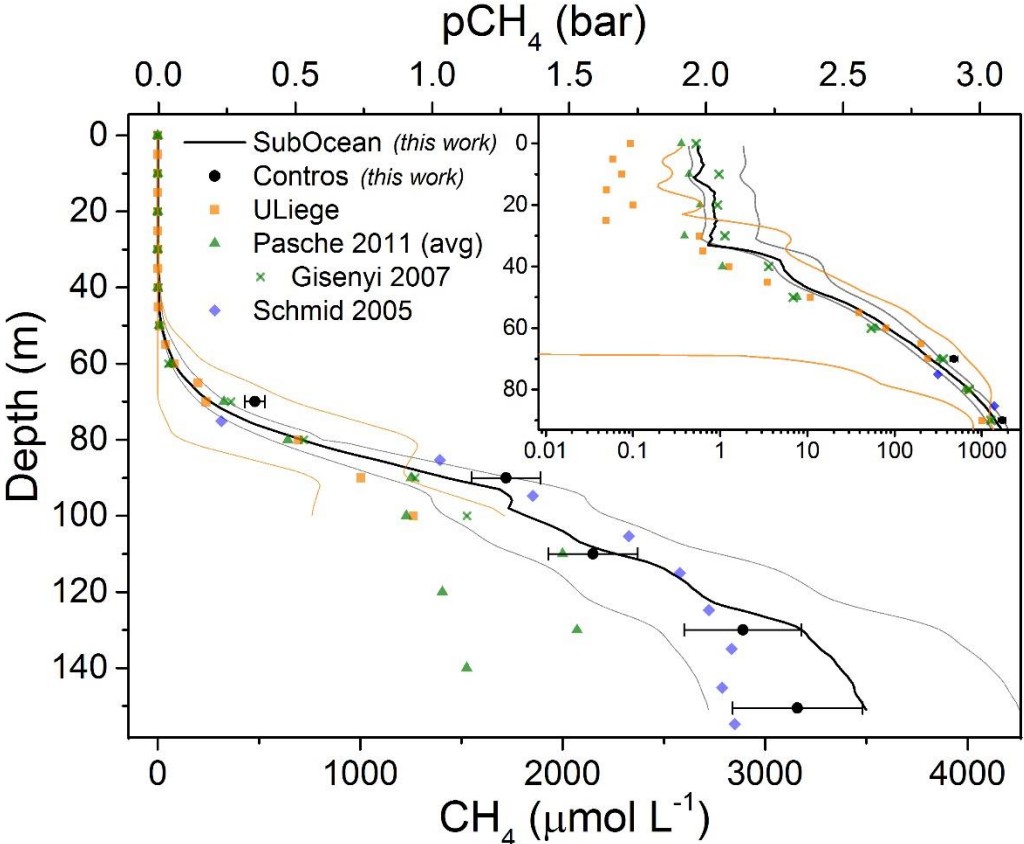

**Figure 6**. Continuous methane profile of the upper 150 m of water depth in Lake Kivu measured by the Sub-Ocean instrument (black line). Grey lines represent the measured variability over the eight continuous profiles estimated between 0 and 80 m depth and fixed to the estimated uncertainty of ± 22 % at larger depths. Black dots are discrete measurements made with the Contros HydroC® HP sensor at different depths. Error bars corresponds to the estimated uncertainty of ± 10 %. Orange squares are from the long term monitoring from the University of Liege (Roland et al., 2017, 2018) with the corresponding 3σ variability (orange lines). Green triangles are average concentrations from Pasche et al. 2011 (Pasche et al., 2011) from three different campaigns conducted in May 2006 and 2007 at different locations (Kibuye, Ishungu and Gisenyi). Green crosses are data from Gisenyi 2007. Blue rhombus correspond to measurements from Schmid et al. 2005 in the northern basin using a commercial Capsum Met sensor (Schmid et al., 2005). In the insert a zoom on the shallow data is presented with a log-scale on the concentrations allowing a better comparison of the different datasets.

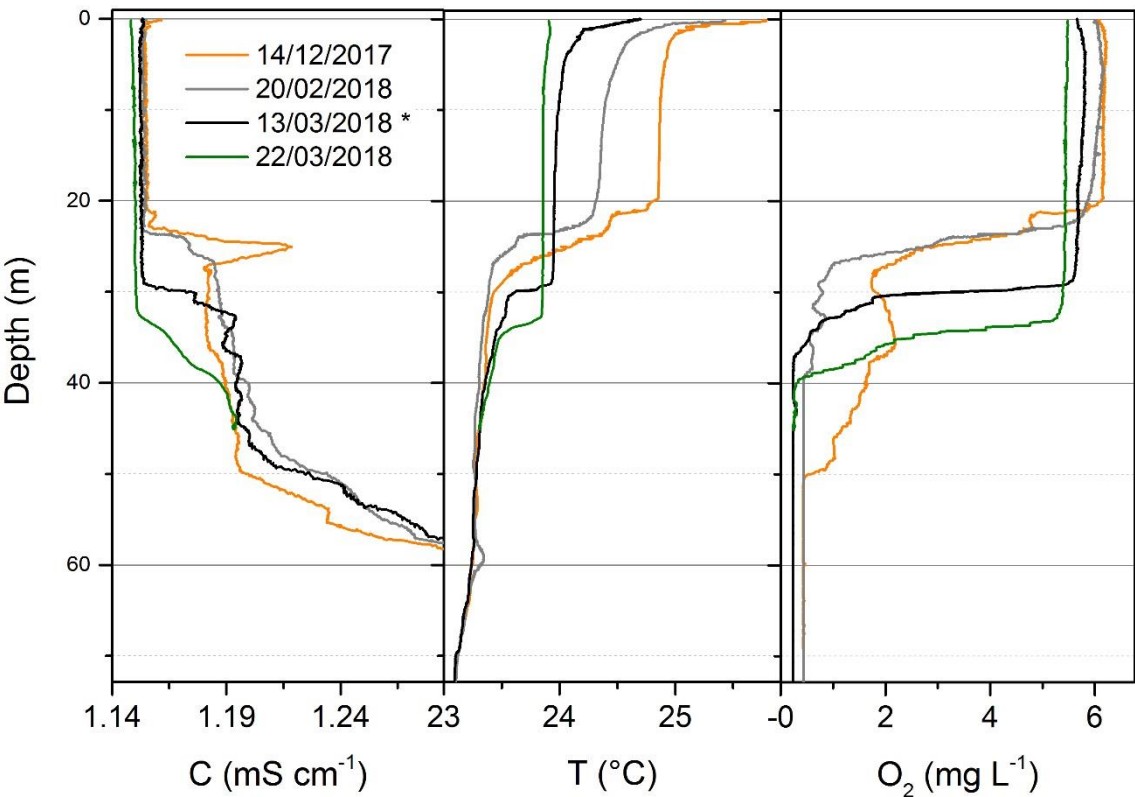

**Figure 7.** CTD (conductivity at 25°C, temperature and dissolved oxygen) data obtained a few months prior to the campaign. The black lines correspond to the conditions during the field measurements (*). The $O_2$ profiles highlight how the mixing layer extended down to 50 m depth during the previous dry season. From mid-December, the lake started to stratify at 25 m, while at the beginning of March the oxic layer increased down to 35 m depth.