# Peer review of "Continuous In Situ Measurement of Dissolved Methane in Lake Kivu Using a Membrane Inlet Laser Spectrometer"

_Geoscientific Instrumentation, Methods and Data Systems, 2019_

## Referee Comment (RC1) · Anonymous Referee #1 · 17 Dec 2019

**Comments on**
**'Continuous In Situ Measurement of Dissolved Methane in Lake Kivu**
**Using a Membrane Inlet Laser Spectrometer' by Grilli et al., GI-2019-29**

The manuscript describes the deployment of a new sensor for the in-situ measurements of dissolved methane concentrations in Lake Kivu, known for its very high concentration of CH4, especially in the deep waters. The sensor is based on a membrane inlet to extract the gas from the water, which is then analysed using a laser spectrometer. This technology has been already described in other papers (Grilli et al 2018 (ES&T), Jansson et al 2019 (OS)) and has proven reliable for low concentration measurements (down to the nM) mainly in oceans.

The challenge here is to evaluate the performance of the technology in very high concentrations (several mmol/l). Lake Kivu offers indeed the perfect conditions for this. It appears that some tuning of the instrument was necessary and yet, the authors concluded that the upper limit for concentration measurement was 3.5 mmol/l even after reducing the sensitivity, while concentrations in the lake can reach 18 mmol/l.

One can see the interest of monitoring the dissolved methane concentrations in Lake Kivu for safety reasons but, it may not be necessary to use a too sensitive technique. As presented, the technique based on laser spectrometer cannot respond to the (important) question of the accumulation rate of CH4 in the deep layers (highlighted in the introduction). Maybe the same technique could be used in a 'high-concentration' mode by using a less sensitive laser spectrometer. Is it possible to use the near IR absorption bands of CH4? i.e. 1.3/1.6 µm? Another way of development could be to reduce the exchange surface of the membrane. I also wonder if going for a Teflon membrane was a good change as Teflon is known for a better permeability to methane (although there are different types of Teflon but the authors do not give any precision on this). In fact, I don't think I properly understood this modification of the instrument. A scheme of the membrane block would help the reader.

Regarding the surface measurements, I would have liked some GC measurements as reference (from samples taken in the same time as MILS measurements). I don't think we can use the data from a commercial sensor as reference.

One thing we can conclude from this deployment is that in situ sensors must be carefully chosen according to the environmental conditions and the scientific question, and adapted accordingly. Because this paper is a good illustration of the constraints of in situ sensor development and also because the technique is promising despite its limitations when deployed in high concentrations, I would recommend its publication in Geoscientific Instrumentation.

However, I have some comments on the form of the manuscript. My main problem is the presentation of the results and the discussion that follows. I would go for a 'results and discussion' section followed by a conclusion instead of the current structure. This would avoid the discussion of some of the results in the results section (line 240-255) and repetition in the Discussion section (line 270). Otherwise, the manuscript is clearly written, figures are well described and clear although I would put the units into brackets, e.g. CH4/% changed to CH4 (%) to avoid any confusion.

Finally, I have only a few typo corrections:

*Line 232: 'therefore not...'* therefore no…

*Line 263: 'not spatial variability...'* no spatial variability

---

## Referee Comment (RC2) · Anonymous Referee #2 · 25 Dec 2019

This paper seeks to demonstrate the reliability of a membrane-based system to continuously measure methane in aquatic environments other than oceans. The authors place their research in stratified lake Kivu, in which water can have very high methane concentrations, and span several orders of magnitude. It is shown how a manipulation of the sensor system can accommodate this large dynamic concentration range, making the presented sensor system a potentially interesting and useful addition to the biogeochemists' toolbox. Such an in-situ sensor system, usable in methane-rich freshwater environments, is of course interesting and relevant, and some of the results are indeed very encouraging, but at the same time the manuscript suffers from some conceptual and structural weaknesses that in my opinion undermine its suitability for

publication in the present form.

General points: Introduction: The authors choose with Lake Kivu an interesting, but also particular case. To me, it is unclear, why the manuscript introduction focusses on lake Kivu (given the choice of journal) instead of focusing on the need for continuous methane profiles in aquatic systems to answer a large array of very interesting and pressing questions, including e.g., transport, production/consumption etc.. While this would require substantial changes to the introduction, it would make a much more useful and strong manuscript. The method presented here is different from a previous paper (Grilli 2018) mostly in the altered measurement mode that allows for much higher concentration to be quantified. That this extended range is still not extensive enough to cover the range in CH4 concentrations of lake Kivu is unfortunate but not critical. A review of the concentration range in freshwater systems would help the reader to understand the relevance of the presented instrument modifications.

Methods: Even though much of the general aspects of the sub-ocean instruments have recently been described (Grilli 2018), the general setup must be presented here in detail. Further, it is unclear to me how the authors arrive at a systematic error of 10% for the HydroC. I am missing information for the use of the instrument (e.g. lowering velocity, frequency of measurement), which should be moved away from the results section.

Results: The authors often mix "discussion" and "results" elements which leads to confusion and unnecessary repetition (up to the point where they mention that the "discussion of a certain value can be found in the results section, L269). Any discussion around the underlying reasons for the measurement uncertainties surely do not belong in the results section. Many of the aspects on lake Kivu methane concentrations are unnecessary, and the comparisons of past CH4 measurements in Kivu through in time and space should be better linked to each other (making clear when technical and when ecological reasons drive differences), and synthesized, of course, in the discussion section. To me, it seems odd to compare the average results measured in the surface

waters with other values that, depending on situation (e.g. season and mixing), span 3 orders of magnitude (L 236).

Discussion: basically missing. Much of the discussion (actually found in the results section) oddly focusses on Kivu-specific observations (e.g. temporal mixing dynamics) instead of methodological aspects. The actual discussion section is mostly a repetition of introduction elements alongside some of the uncertainty numbers mentioned in the results. The latter were derived based on the assumption of homogenous methane concentration across the lake (L263), although the distance from the sediments is known to have large influence on local methane concentrations (del Sontro 2018). While this observation is mostly true for epilimnetic waters, it is unclear how if the hydrodynamic features of Lake Kivu allows for omission of this important control of the spatial distribution of CH4 in lakes. Further, as a reader or potential user of such an instrument, I'd be highly interested in performace metrics (e.g. opposed to other techniques and instruments), or reasons for particular performance-related issues (e.g. regarding the carrier gas flow), and these aspects therefore require much more elaboration.

Specific points: Figure 5: panel TDGP, point 70-90m, how can there be a nonlinearity in the interpolation? What value is interpolated?

Figure 6: I understand the quantification of $\sigma$ for precision and repeatability. However, it seems overly simplified to express the error quantified at one concentration in percent, and then extrapolate the relative error across all orders of magnitude (making very small absolute errors in small concentration).

L90: It would be interesting to know how long the instrument could run with "between 2 and 40 bar" of carrier gas.

L185: general information about complementary sampling campaign and their published results sshould be shortened

L214 N2 calculations from other measured gases is used in a figure and should therefore be part of the method section.

L221 Orange lines in what figure?

L232 I don't understand this piece of information

Technical corrections

Sloppiness with the references does not increase the joy of reading this manuscript.

I am no native English speaker myself, but many terms and expressions seem awkward (e.g., L86 "at the price of", L118 "electromechanical cable", L176, L191 "than", L236 "higher edge", L261 "seeing a background" ).

Typos

L246 "O2 completely vanished"

L249 reasons

L244 e.g.

L433 replace "retrieved"

---

## Author Comment (AC1) · 14 Jan 2020

We thank the expert reviewer for the very fruitful comments and remarks, which helped us to improve the manuscript.

All the remarks been addressed below, and changes in the manuscript have been done accordingly using word track changes. Some grammatical improvements have also been done. The instruments have been named more consistently over the manuscript as SubOcean and HydroC HP.

We hope that the manuscript is now acceptable for publication in Geoscientific Instrumentation, Methods and Data Systems.

Reviewer(s)' Comments to Author:

The manuscript describes the deployment of a new sensor for the in-situ measurements of dissolved methane concentrations in Lake Kivu, known for its very high concentration of CH4, especially in the deep waters. The sensor is based on a membrane inlet to extract the gas from the water, which is then analysed using a laser spectrometer. This technology has been already described in other papers (Grilli et al 2018 (ES&T), Jansson et al 2019 (OS)) and has proven reliable for low concentration measurements (down to the nM) mainly in oceans.
The challenge here is to evaluate the performance of the technology in very high concentrations (several mmol/l). Lake Kivu offers indeed the perfect conditions for this. It appears that some tuning of the instrument was necessary and yet, the authors concluded that the upper limit for concentration measurement was 3.5 mmol/l even after reducing the sensitivity, while concentrations in the lake can reach 18 mmol/l.
One can see the interest of monitoring the dissolved methane concentrations in Lake Kivu for safety reasons but, it may not be necessary to use a too sensitive technique. As presented, the technique based on laser spectrometer cannot respond to the (important) question of the accumulation rate of CH4 in the deep layers (highlighted in the introduction). Maybe the same technique could be used in a 'high-concentration' mode by using a less sensitive laser spectrometer. Is it possible to use the near IR absorption bands of CH4? i.e. 1.3/1.6 µm?

Thanks for this interesting and pertinent remarks. As mentioned in the manuscript, we agree with the reviewer that the high sensitive spectrometer working at 2.3 µm (generally used for trace gas detection) is not well adapted for the measurement in Lake Kivu. Moving to 1.6 or even 1.3 µm could indeed help to decrease the sensitivity by one and two order of magnitude, respectively, but it would have required a dedicated development and it this case it would have been preferable to switch to a less complex technique based on direct or multipass absorption probing strong fundamental transitions. The spectrometer used in this work was developed for study the fate of methane in the ocean (see the work Jansson et al 2019 (OS)) for which the high sensitivity was a key feature. The campaign at Lake Kivu was for us more an opportunity to test the adaptation of our extraction technique (which provides fast response time) in a very different and harsh environment (high dissolved gas concentrations and pressures, anoxic environment…). With this work we proved that the extraction technique is well suitable for such environments, and that in the future in situ sensors based on this technique could be used for fast monitoring of those type of lakes.

Another way of development could be to reduce the exchange surface of the membrane. I also wonder if going for a Teflon membrane was a good change as Teflon is known for a better permeability to methane (although there are different types of Teflon but the authors do not give

any precision on this). In fact, I don't think I properly understood this modification of the instrument. A scheme of the membrane block would help the reader.

The membranes used here are made in PDMS. This is now specified in the manuscript (line 90). A drawing of the membrane block was reported in the SI of Grilli et al 2018, and therefore not further added in this work. We decreased already the membrane surface by replacing one of the two membranes with a gas-tight Teflon film (this is now more clear in the text (line 110)).

Regarding the surface measurements, I would have liked some GC measurements as reference (from samples taken in the same time as MILS measurements). I don't think we can use the data from a commercial sensor as reference.

The reviewer is correct, GC measurements at shallower depths would have been very profitable, but unfortunately this was not done during the campaign. However, two other techniques were used at depths below 150 m: an "on site" mass spectrometer sensor analyzed water pumped from different depths developed at EAWAG (Switzeland) and a water sampling followed by GC measurements performed by the group at UFZ (Germany) (Boehrer et al, HESS 2019). The HydroC commercial instrument well agreed with those other two methods up to a depth of 250 m, afterwards a discrepancy was observed with a systematic underestimation of 12% by the HydroC sensor. Further details can be founded in the official report of the campaign Schmid et al, 2019. Here, an extract of the figure presented in the report (section 3.5 showing an intercomparison of the four techniques used during the campaign). Red cicles corresponds to the HydroC HP measurements, while the green line corresponds to our continuous measurements.

[Figure]

*Figure 3.5: Summary of methane concentrations measured in the intercalibration campaign.*

This discussion has now been added in the manuscript:

« During the campaign the HydroC HP sensor also showed a good agreement with the other discrete techniques (on site mass spectroscopy and discrete sampling followed by GC analysis) between 150 and 250 m, while at larger depths, the HydroC HP values were lower by ~12% (Schmid et al., 2019). »

One thing we can conclude from this deployment is that in situ sensors must be carefully chosen according to the environmental conditions and the scientific question, and adapted accordingly. Because this paper is a good illustration of the constraints of in situ sensor development and also because the technique is promising despite its limitations when deployed in high concentrations, I would recommend its publication in Geoscientific Instrumentation. However, I have some comments on the form of the manuscript. My main problem is the presentation of the results and the discussion that follows. I would go for a 'results and discussion' section followed by a conclusion instead of the current structure. This would avoid the discussion of some of the results in the results section (line 240-255) and repetition in the Discussion section (line 270). Otherwise, the manuscript is clearly written, figures are well described and clear although I would put the units into brackets, e.g. CH4/% changed to CH4 (%) to avoid any confusion.

We thank the reviewer for supporting the publication of our work. The organization of the manuscript and the units have been changed accordingly. Thanks for these suggestions which improved the structure of the manuscript.

Finally, I have only a few typo corrections:

Line 232: 'therefore not…' therefore no…

Corrected

Line 263: 'not spatial variability…' no spatial variability

Corrected

---

## Author Comment (AC2) · 14 Jan 2020

We thank the expert reviewer for the very fruitful comments and remarks, which helped us to improve the manuscript.

All the remarks been addressed below, and changes in the manuscript have been done accordingly using word track changes. Some grammatical improvements have also been done. The instruments have been named more consistently over the manuscript as SubOcean and HydroC HP.

We hope that the manuscript is now acceptable for publication in Geoscientific Instrumentation, Methods and Data Systems.

Reviewer(s)' Comments to Author:

This paper seeks to demonstrate the reliability of a membrane-based system to continuously measure methane in aquatic environments other than oceans.  The authors place their research in stratified lake Kivu, in which water can have very high methane concentrations, and span several orders of magnitude.  It is shown how a manipulation of the sensor system can accommodate this large dynamic concentration range, making the presented sensor system a potentially interesting and useful addition to the biogeochemists' toolbox.  Such an in-situ sensor system, usable in methane-rich freshwater environments, is of course interesting and relevant, and some of the results are indeed very encouraging, but at the same time the manuscript suffers from some conceptual and structural weaknesses that in my opinion undermine its suitability for publication in the present form.

General points:

Introduction:  The authors choose with Lake Kivu an interesting, but also particular case. To me, it is unclear, why the manuscript introduction focusses on lake Kivu (given the choice of journal) instead of focusing on the need for continuous methane profiles in aquatic systems to answer a large array of very interesting and pressing questions, including e.g., transport, production/consumption etc.. While this would require substantial changes to the introduction, it would make a much more useful and strong manuscript. The method presented here is different from a previous paper (Grilli 2018) mostly in the altered measurement mode that allows for much higher concentration to be quantified.  That this extended range is still not extensive enough to cover the range in CH4 concentrations of lake Kivu is unfortunate but not critical. A review of the concentration range in freshwater systems would help the reader to understand the relevance of the presented instrument modifications.

We thank the reviewer for those remarks. The introduction has been adjusted as suggested. The importance of better constraint the CH4 emissions to the atmosphere and the processes behind those emissions in aquatic environments is now discussed. We did not reduce the information on Lake Kivu since, even if the paper is centered on the measurement technique, we found important to provide the contest to the reader, and to highlight the importance of the dissolved gas monitoring at this particular site. Safety reasons are driving this monitoring of the evolution of the lake, particularly now that power plants are exploiting the gas reservoir. The reviewer also asked a review of concentration ranges in freshwater systems. We are unfortunately not in the position of providing an extensive review, but we report two examples of meromictic lakes where the upper measurement range of the Sub-Ocean probe is compatible with the concentration of bottom waters. We hope that the reviewer will now consider this introduction better adapted to the focus of this work.

Methods:  Even though much of the general aspects of the sub-ocean instruments have recently been described (Grilli 2018), the general setup must be presented here in detail. Further, it is unclear to me how the authors arrive at a systematic error of 10% for the HydroC. I am missing information for the use of the instrument (e.g. lowering velocity, frequency of measurement), which should be moved away from the results section.

The information about the lowering speed, frequency of the acquisition and response time for the SubOcean instrument have now been added in the Method section.

"The embedded spectrometer is continuously measuring the gas composition at 10 Hz, while the response time of the sensor during the campaign, expressed as $\tau_{90}$, was ~10 sec. At a lowering speed of ~6 m min$^{-1}$, this corresponds to a vertical resolution of 1 m".

The 10% for the HydroC was specified by the manufacturer though a personal communication.

Further information on the extraction system has now been added in the manuscript.

"A stainless-steel membrane block (MB) was equipped with two 10 µm thick polydimethylsiloxane (PDMS) membranes of 56 mm diameter mounted face-to-face. The membranes were mounted on porous bronze frits of 3 mm of thickness (Poral, grade 20), providing mechanical strength for the membrane under high pressure differences. A schematic of the membrane block can be found in the supplementary information of Grilli et al., 2018."

Results:  The authors often mix "discussion" and "results" elements which leads to confusion and unnecessary repetition (up to the point where they mention that the "discussion of a certain value can be found in the results section, L269). Any discussion around the underlying reasons for the measurement uncertainties surely do not belong in the results section.  Many of the aspects on lake Kivu methane concentrations are unnecessary, and the comparisons of past CH4 measurements in Kivu through in time and space should be better linked to each other (making clear when technical and when ecological reasons drive differences), and synthesized, of course, in the discussion section. To me, it seems odd to compare the average results measured in the surface waters with other values that, depending on situation (e.g. season and mixing), span 3 orders of magnitude (L 236).

Indeed the comparison on surface measurements is not easy to do due to seasonality and mixing. The reviewer is right saying that the span of existing data is large, but we noticed that 1) our measurement is compatible with previous ones; 2) that high values (as high as our) were previously measured on this time of the year and reported by Pasche et al 2011; 3) The CDT data from few months prior the campaign highlighted a mixing event occurring at the beginning of March which would explain this large amount of CH4 at the surface. Even if we do not know what cause this mixing we could clearly observe it from the data reported in Figure 7. From the O2 profiles one can notice that from mid-December the lake started to stratify at 25 m (bringing water enriched in dissolved gas up to this depth), while at the beginning of March the oxic layer increased down to 35 m depth, highlighting a mixing of this upper part which would cause an increase in dissolved gas at the surface.

We now improved the structure, making a Results and Discussion section followed by a conclusion section.

Discussion:  basically missing.  Much of the discussion (actually found in the results section) oddly focusses on Kivu-specific observations (e.g.  temporal mixing dynamics) instead of methodological aspects.  The actual discussion section is mostly a repetition of introduction elements alongside some of the uncertainty numbers mentioned in the results. The latter were derived based on the assumption of homogenous methane concentration across the lake (L263), although the distance from the sediments is known to have large influence on local methane concentrations (del Sontro 2018). While this observation is mostly true for epilimnetic waters, it is unclear how if the hydrodynamic features of Lake Kivu allows for omission of this important control of the spatial distribution of CH4 in lakes. Further, as a reader or potential user of such an instrument, I'd be highly interested in performace metrics (e.g.  opposed to other techniques and instruments), or reasons for particular performance-related issues (e.g.  regarding the carrier gas flow), and these aspects therefore require much more elaboration.

We thanks the reviewers for this remark and input. The reviewer is right saying that concentration of dissolved gas are not constant over the whole surface of the lake, and that could strongly depends on the depth (i.e. distance of the sediments to the surface). With the SubOcean instrument we recently conduct two field campaigns, one in the Black Sea and another one at Lake Aiguebelette (France) that both highlighted this behavior. We therefore added in the discussion the following part:

"This fast response sensor could be used to better investigate the fluxes of CH4 (or other greenhouse gases) from lakes, oceans, rivers and other water reservoirs. In this campaign only a specific location at 5 km from the coast with 410 m of water depth was investigated. The amount of CH4 at the surface could strongly depends on the water depth i.e. on the distance of the sediment to the surface. A fast sensor could allow to follow the spatial distribution of dissolved gas at the surface for different depths, as well as its variability over the seasons. This would help to better constrain the greenhouse gas emissions in the face of global change (DelSontro et al., 2018)."

The Discussion have now been merged with the Results section as suggested by both reviewers. We hope that the actual structure make the manuscript clearer.

Specific points:

Figure 5: panel TDGP, point 70-90m, how can there be a nonlinearity in the interpolation? What value is interpolated?

This has now been corrected. Thanks for spotting this out.

Figure 6: I understand the quantification of σ for precision and repeatability. However, it seems overly simplified to express the error quantified at one concentration in percent, and then extrapolate the relative error across all orders of magnitude (making very small absolute errors in small concentration).

This remark is indeed correct, but it is challenging to report the accuracy of the sensors in the upper layers where the lake is not stratified. We therefore decided to report the actual variability of the measurements at depths from 0 to 80 m, and to fix the error bars to the value estimated at 80 m at lower depth (since only one profile cover the 150 meters, and all the others saturated within the first 100 m). We added this discussion in the text and adapter the Figure accordingly.

« The uncertainty represented by the grey lines in **Erreur ! Source du renvoi introuvable.** represents the measured variability over the eights vertical profiles from 0 to 80 m, and was fixed to ± 22% at larger depths »

L90: It would be interesting to know how long the instrument could run with "between 2 and 40 bar" of carrier gas.

Different aspects must be considered for providing the maximum long-term deployment of the instrument. The carrier gas flow may be varied between 0 and 6 ml/min STP. The gas that is analyzed is stored inside the instrument. Since no miniaturization efforts were made for this prototype, the internal free volume is relatively large, estimated to 30 L, which allows to store a continuous flow of 6 ml/min of gas for 24h, before the pressure reaches 1.3 bar (absolute pressure) inside the instrument that would make the embedded vacuum pump no longer capable to provide the 30 mbar inside the measurement cell. The 40 bar reported in the manuscript represent a large excess than what is actually required, even with large carrier gas flows (with 6 ml/min STP of carrier gas 10 bar in 1L tank will already be enough for the 24h measurements). But for precaution, the tank is normally filled with a larger amount of carrier gas than what is required (eg. in case of small leaks in the high-pressure part of the line).

At low carrier gas flow, the amount of dry gas is very small, the mixture is mainly composed by water vapor which is trapped with silica gel before reaching the pump. So in the ideal case where all the water vapor is trapped and that the only gas reaching the pump is the dissolved dry gas (0.065 ml/min STP) then the autonomy of the instrument will be 90 days. This discussion has now been added on the manuscript:

"For an operation where the instrument is powered through an electromechanical cable the autonomy will be limited by the storage of the dry gas inside the instrument housing. For fast response measurements at maximum carrier gas flow of 6 ml min$^{-1}$ this will correspond to 24 h autonomy, whereas without the use of carrier gas the autonomy will stretch to 90 days since most of the gas flow will be composed of water vapor that is trapped before the vacuum pump by the silica gel dryer (however, the long-term deployment may be limited by the capability of the silica gel)."

L185: general information about complementary sampling campaign and their published results should be shortened

Sorry, but we found difficult to make it shorter.

L214 N2 calculations from other measured gases is used in a figure and should therefore be part of the method section.

Changed accordingly

L221 Orange lines in what figure?

Figure 6. The information has been added.

L232 I don't understand this piece of information

We tried to make it clearer. The data at higher depths (>150 m) measured with the Contros HydroC sensor were about 12% lower with respect to the measurements by EAWAG and UFZ. We think that this is possibly due to the calibration of the sensor by the manufacturer but we do not have further information which could confirm this hypothesis.

Technical corrections

Sloppiness with the references does not increase the joy of reading this manuscript.

We are sorry to hear that the reviewer found missing references. But without further details on this remark we are not able to improve the manuscript accordingly.

I am no native English speaker myself, but many terms and expressions seem awkward (e.g., L86 "at the price of", L118 "electromechanical cable", L176, L191 "than", L236 "higher edge", L261 "seeing a background" ).

Those expressions have been changed/corrected (a part for "electromechanical cable and "higher edge" that we find correctly used).

Typos
L246 "O2 completely vanished"
L249 reasons
L244 e.g.

L433 replace "retrieved"

Typos errors have been corrected as suggested.

---

## Referee Report (RR1)

**Comments on**
**'Continuous In Situ Measurement of Dissolved Methane in Lake Kivu**
**Using a Membrane Inlet Laser Spectrometer' by Grilli et al., GI-2019-29**

**Revised submission**

I thank the authors for re-submitting their paper considering our comments as well as answering our questions. Most of my comments were addressed, especially the 'results and discussion' section, which is now much easier to read.

I still think this paper is worth a publication as it brings a new in situ technique for the measurements of an important greenhouse gas in aquatic environments. The scarcity of data from these specific systems is clearly a limitation for constraining the budgets and therefore the models, so any advance in the field of sensor development must be encouraged.

However, I have noticed many typos that make the MS difficult to read, which should not happen with a re-submission. Also, some re-phrasing would make the text clearer. I suggest a careful review from the authors to avoid these mistakes.

Here are my detailed comments:

Line 26: 'Methane (CH4)'… putting (CH4) there would avoid to do it line 43
Line 35: costal -> coastal
Line 35-37: 'a better understanding of the processes…  is  required'. But I'd re-phrase as follows 'Only fast response instrument for in situ dissolved gas measurements and dynamic profiling can provide the data for a better understanding of the processes…'
Line 40: 'water' is too many times used, so I'd suggest: '… makes deep water strongly decoupled from the surface layer because…'
Line 41: delete 'and therefore very different in composition'
Line 43: delete methane '… dissolved carbon dioxide (CO2) and CH4…'
Line 45-46: check the brackets '… present in the lake, e.g. Degens et al. (1973), Pasche et al. (2011)…' Please be careful to the way the references are cited. This should be homogenized throughout the MS.
Line 47: same comment about the citations
Line 53: 'Regarding the stability of the lake, Schmid et al. (2005) raised…'
Line 59: content -> concentration
Line 70: 'highlighted in the discussion section, in comparison with other methods deployed during the same campaign: water sampling followed…'
Line 74-77: Presented this way, this should not be in the introduction but in the discussion section. If the authors want to follow the 2[nd] reviewer's comments, then this should be higher in the introduction, but I still think this should be given as comparison in the discussion.
Line 85: PDMS is for polydimethylsiloxane. It should read 'using a PolyDiMethylSiloxane (PDMS) membrane…'
Line 92: check the brackets '…can be found in Grilli et al. (2018).'
Line 99: delete polydimethylsiloxane as the acronym is already clarified line 85
Line 102: again, check the brackets for the reference '… Grilli et al. (2018)'
Line 119: dot missing -> 'speed of ~6m.min$^{-1}$'
Line 120: space missing 'after resolution of 1m.'
Line 126: HydroC is a registered trademark so this should be noted as '… underwater sensor, the CONTROS HydroC® HP sensor'. This should be modified throughout the MS.

Line 129: same remark as above regarding the registered trademark. Also, another problem with the brackets, which should be 'in Fietzek et al. (2014), …'.

Line 181: Sander 2015 cited twice… this should read '… equation 19 from Sander (2015) using…'. Delete (Sander, 2015) at the end.

Line 197: reference cited twice!

Line 203: same remark

Line 219: but not at the same depths!

Line 239: delete (Schmid et al. 2015) at the end (already in the sentence).

Line 241: 'Nitrogen (N2) mixing ratio…'

Line 251-252: Brackets missing

Line 253: the headspace technique is an extraction technique. The analysis is done using Gas Chromatography.

Line 256: Brackets missing

Line 260: greater instead of larger

Line 265: are the unpublished data from Roland et al.?

Line 266: brackets missing

Line 275: 'O2 supplied at these depths during the previous dry season  completely vanished'

Line 282: brackets missing

Line 288: I think the amount of CH4 in the surface layer mainly depends on the biogeochemical processes, especially in presence of oxygen. To name one: bacterial oxidation of methane. Yes there is a dilution of CH4 from the anoxic layer to the oxic one but methanotrophy is the main process that control the concentration in any kind of aquatic environments.

Line 328: replace 'discrete sampling' by 'discrete measurements' to avoid confusion between the water sampling done with the Niskin bottles and the discrete in situ measurement performed with the sensor. Closing a Niskin bottle takes less than 1 second, so a 410m profile can be done in less than 10 min…

---

## Author Response (AR2)

**Referee#1**

**Comments on**
**'Continuous In Situ Measurement of Dissolved Methane in Lake Kivu**
**Using a Membrane Inlet Laser Spectrometer' by Grilli et al., GI-2019-29**
**Revised submission**

I thank the authors for re-submitting their paper considering our comments as well as answering our questions. Most of my comments were addressed, especially the 'results and discussion' section, which is now much easier to read.

I still think this paper is worth a publication as it brings a new in situ technique for the measurements of an important greenhouse gas in aquatic environments. The scarcity of data from these specific systems is clearly a limitation for constraining the budgets and therefore the models, so any advance in the field of sensor development must be encouraged. However, I have noticed many typos that make the MS difficult to read, which should not happen with a re-submission. Also, some re-phrasing would make the text clearer. I suggest a careful review from the authors to avoid these mistakes.

We thank the referee for the critical review. We addressed all the comments below and we carefully checked the manuscripts for other typo and grammatical errors. We also rearranged some sentences for making the manuscript easier to read.

Here are my detailed comments:

Line 26: 'Methane (CH4)'… putting (CH4) there would avoid to do it line 43
Corrected

Line 35: costal -> coastal
Corrected

Line 35-37: 'a better understanding of the processes… are is needed required'. But I'd re-phrase as follows 'Only fast response instrument for in situ dissolved gas measurements and dynamic profiling can provide the data for a better understanding of the processes…'
Corrected accordingly

Line 40: 'water' is too many times used, so I'd suggest: '… makes deep water strongly decoupled from the surface layer because…'
Modified as suggested

Line 41: delete 'and therefore very different in composition'

We changed but we kept to word composition in the sentence: "because of their difference in density and composition".

Line 43: delete methane '… dissolved carbon dioxide (CO2) and CH4…'
Corrected

Line 45-46: check the brackets '… present in the lake, e.g. Degens et al. (1973), Pasche et al. (2011)…' Please be careful to the way the references are cited. This should be homogenized

throughout the MS.
Line 47: same comment about the citations

Regarding citations we used the Mendeley citation style for GI journal where the year should not be under brackets. We therefore leave the citation format as it is.

Line 53: 'Regarding the stability of the lake, Schmid et al. (2005) raised…'

We decided to write "in 2005 Schmid and co-workers raised" and move the citations to the end of the sentence.

Line 59: content -> concentration
Corrected

Line 70: 'highlighted in the discussion section, in comparison with other methods deployed during the same campaign: water sampling followed…'
Corrected

Line 74-77: Presented this way, this should not be in the introduction but in the discussion section. If the authors want to follow the 2nd reviewer's comments, then this should be higher in the introduction, but I still think this should be given as comparison in the discussion.
We agreed with the referee that this part would better fit in the discussion session. And it has been moved accordingly.

Line 85: PDMS is for polydimethylsiloxane. It should read 'using a PolyDiMethylSiloxane (PDMS) membrane…'
Corrected

Line 92: check the brackets '…can be found in Grilli et al. (2018).'
Corrected

Line 99: delete polydimethylsiloxane as the acronym is already clarified line 85
Corrected

Line 102: again, check the brackets for the reference '… Grilli et al. (2018)'
Corrected

Line 119: dot missing -> 'speed of ~6m.min-1'
Corrected

Line 120: space missing 'after resolution of 1m.'
Corrected

Line 126: HydroC is a registered trademark so this should be noted as '… underwater sensor, the CONTROS HydroC® HP sensor'. This should be modified throughout the MS.
Corrected

Line 129: same remark as above regarding the registered trademark. Also, another problem with the brackets, which should be 'in Fietzek et al. (2014), …'.
Line 181: Sander 2015 cited twice… this should read '… equation 19 from Sander (2015) using…'. Delete (Sander, 2015) at the end.
Citations have been revised

Line 197: reference cited twice!
Corrected

Line 203: same remark
Corrected

Line 219: but not at the same depths!
Now mentioned "and over specific discrete depths."

Line 239: delete (Schmid et al. 2015) at the end (already in the sentence).
Corrected

Line 241: 'Nitrogen (N2) mixing ratio…'
Corrected

Line 251-252: Brackets missing
Line 253: the headspace technique is an extraction technique. The analysis is done using Gas Chromatography.
Modified accordingly

Line 256: Brackets missing
Line 260: greater instead of larger
Corrected

Line 265: are the unpublished data from Roland et al.?
Yes. This is now clearly mentioned

Line 266: brackets missing
We could not see the missing brackets highlighted by the referee.

Line 275: 'O2 supplied at these depths during the previous dry season was completely vanished'
No clear: this is the phrase as appear in the manuscript. What should be changed here?

Line 282: brackets missing
Line 288: I think the amount of CH4 in the surface layer mainly depends on the biogeochemical processes, especially in presence of oxygen. To name one: bacterial oxidation of methane. Yes there is a dilution of CH4 from the anoxic layer to the oxic one but methanotrophy is the main process that control the concentration in any kind of aquatic environments.
We now added the sentence: "Finally, CH4 concentration in the surface layer may depend on the biogeochemical processes such as for instance the methanotrophy."

Line 328: replace 'discrete sampling' by 'discrete measurements' to avoid confusion between the water sampling done with the Niskin bottles and the discrete in situ measurement performed with the sensor. Closing a Niskin bottle takes less than 1 second, so a 410m profile can be done in less than 10 min…
Changed.

**Referee#2**

The manuscript has improved by the revision, the necessary context is now included, the new structure is in somewhat better shape, and the technical details are presented in sufficient depths. Still, I find that the authors didn't show a satisfactory effort in condensing the manuscript, which is therefore quite long, presents a range of unrelated information bits, and both aspects make it difficult to read. For a manuscript like this, lacking a testable biogeochemical hypothesis, to warrant publication it is necessary that all elements are clearly structured and the (necessary) information is complete and easily accessible. In my opinion, the combination of results and discussion section is not ideal, and I cannot recommend the manuscript in its present form for publication if the scale is clarity. If Geoscientific Instrumentation .. is happy with the presented kind of report it would be acceptable, though, after some specific points were addressed.

We are sorry to hear that the referee found the manuscript not easy to read. We think that the information provided are complete and easy to access. We think that some generic information about the lake and the methane extraction are important to understand the motivation for comparing different techniques and for a more accurate estimation of the methane budget of the lake. We made further efforts in order to improve the English and more in general the form of the manuscript. We hope that the current version of the manuscript is clear enough and considered acceptable for publication. We thanks again the referee for the very profitable and pertinent remarks that helped us to improve the manuscript.

Specific points.

L26-30 Usually such statements would be accompanied by one or more references.
added

L33 replace "incomes" with "fluxes"
replaced

L42 There is of course also "life" in the anoxic zone (bacteria, archaea,…)
removed from the sentence

L69-70 omit this sentence
Changed according to referee #1

L287 While there is some relation between water depth and methane concentration, the critical point I wanted the authors to make is the horizontal distribution, which is dependent on the distance from the LITTORAL sediments. The title of the reference I was suggesting says it all "No Longer a Paradox: The Interaction Between Physical Transport and Biological Processes Explains the Spatial Distribution of Surface Water Methane Within and Across Lakes", del Sontro 2018. Revise accordingly.
Changed accordingly. "The amount of CH4 at the surface may strongly depend on the water depth, i.e. on the distance of the sediment to the surface, as well as to the horizontal distance from the shore and littoral sediments (DelSontro et al., 2018b)"

If the journal's copy editing isn't very strong, the manuscript will require thorough checking by a native speaker. Some examples

L46 constitutes
Corrected

L49 lake not Lake
Corrected

L55 hypothesis excluded? temporal variability slower?
This seems to be correct for us.

L97 a word seems missing
adjusted with "is no longer strong enough…"

L279 indicate neither…nor
Corrected

L297 "for a correctly determine"
Corrected

Further improvements have been done for making the manuscript clearer and easier to read. We hope that this work can now be accepted for publication.

[revised manuscript text omitted]